# CROSS-REFLECT: EMPOWERING MULTI-MODAL AGENTS WITH JOINT REASONING ACROSS TRAJECTORIES

## ABSTRACT

Despite rapid progress in vision–language models (VLMs), small VLMs still struggle to serve as effective agents capable of coherent multi-step tool use, especially in settings where fine-tuning is impractical due to data or cost constraints. To address these limitations, we introduce Cross-Reflect, a training-free framework for reflection-guided trajectory optimization. Cross-Reflect generates multiple candidate trajectories, applies structured reflection to critique and refine them, and performs cross-trajectory selection to identify the most reliable solution. It is instantiated via an extension of the DSPy programming paradigm, which provides modular support for multimodal inputs. Extensive experiments across static and dynamic knowledge-intensive VQA benchmarks demonstrate that Cross-Reflect consistently improves small VLMs by enabling flexible tool usage and trajectory-level self-reflection, achieving average relative improvements of 10.5% for proprietary models and 28.1% for open-source models over baseline methods. Further analysis shows that our approach achieves comparable performance to methods requiring fine-tuning, and even surpasses them in certain cases.

## 1 INTRODUCTION

Recent advances in Vision Language Models (VLMs) have demonstrated strong instruction-following capabilities (Achiam et al., 2023; Li et al., 2024a; Zhu et al., 2024; Chen et al., 2023b; Bai et al., 2025; Chen et al., 2024), enabling researchers to explore effective and structured ways to use these models (Chiang et al., 2024; Hurst et al., 2024). However, despite these promising developments, small VLMs remain largely unreliable when deployed as autonomous agents. Yet their low inference cost makes them particularly appealing for large-scale or on-device deployment. They often struggle with multi-step tool usage, failing to select the most appropriate tools or construct effective plans throughout their reasoning trajectories. Existing approaches, such as using Process Reward Models (PRMs) or Observation Reward Models (ORMs) (Lightman et al., 2023) to guide decision-making at each reasoning step, have attempted to address this issue. While promising, reward models often lack universality, as they require task-specific training and rely heavily on curated datasets, making them difficult to generalize across diverse tasks. These challenges highlight the need for more robust strategies to enhance the reliability and adaptability of small VLM agents without extensive training.

In this paper, we propose Cross-Reflect, a training-free inference-time framework that enhances the reliability of small VLM agents through reflection-guided trajectory optimization. Unlike prior reflection methods that focus only on final answers or chain-of-thoughts, Cross-Reflect performs structured reflection at the trajectory level: the agent generates multiple candidate reasoning paths, critiques their overall coherence and tool-use strategy, and then selects the most reliable solution through cross-trajectory evaluation. This design allows the agent to improve its reasoning holistically, optimizing the quality of entire trajectories rather than individual steps or components.

To achieve this, we first modularize the generation pipeline vision-language models following DSPy (Khattab et al., 2024), and then build up an agent to call external tools with the modularized VLMs. The agent is initialized with offline prompt optimization. Such constructions allow us to use a training-free model to plan the tool use with minimum manual prompting labor and provide flexibility for further plan optimization. Building upon this, we introduce a structured prompting

mechanism to arouse the agent's self-reflection. After executing each planned reasoning process, the agent reviews its own step-wise decisions and gives structured reflections on them, including tool usage order, utility, and logical consistency. Then it leverages the generated reflections to improve subsequent pipeline construction and reasoning quality iteratively.

We empirically evaluate our framework with both open-source and proprietary VLMs, demonstrating its superior performance and generality across different model families. Experiments on knowledge-intensive VQA show that our Cross-Reflect consistently and substantially improves performance: it achieves average relative accuracy gains of 10.5% for proprietary models and 28.1% for open-source models over their respective baselines. Further ablation studies confirm the importance of reflection-guided trajectory optimization in enhancing both tool selection and reasoning quality.

Our main contributions are summarized as follows:

- We introduce a modular multimodal agent framework by extending the DSPy paradigm to vision-language tasks. The framework abstracts agent behavior into structured modules, enabling flexible and reusable pipeline construction without reliance on handcrafted prompt templates or fine-tuning.
- We propose a reflection-guided trajectory optimization mechanism that evaluates entire trajectories, generates structured reflections, and guides subsequent rollouts. Together with a cross-trajectory selection step, this mechanism transforms reflection into a general test-time optimization principle for multimodal agents.
- Extensive experiments across static and dynamic knowledge-intensive benchmarks demonstrate that our approach achieves state-of-the-art performance among training-free methods, and performs comparably to existing training-based agents, establishing a new strong baseline for knowledge-intensive multimodal question answering.

## 2 RELATED WORKS

**Large model Programming.** Large model programming can be broadly divided into two paradigms. The first paradigm focuses on translating natural language instructions into executable forms, such as Python code or structured expert model calls, enabling large models to perform complex tasks through explicit, interpretable program synthesis. Some works (Beurer-Kellner et al., 2023; Gao et al., 2023; Paranjape et al., 2023; Dong et al., 2024) focus on reasoning and programming in a pure language setting and others are early examples of vision-language programmatic reasoning agents (Gupta & Kembhavi, 2023; Surís et al., 2023). This approach leverages the reasoning and compositional abilities of LLMs to bridge natural language and actionable operations, supporting flexible integration with external tools and models. The second paradigm draws inspiration from deep learning frameworks like Torch (Collobert et al., 2002), aiming to abstract the process of prompting itself. By introducing programming abstractions for LLM pipelines (Khattab et al., 2024), these methods enable users to systematically compose, optimize, and reuse modular prompt components. This emerging foundation model programming framework not only improves scalability and maintainability, but also facilitates more transparent and customizable control over complex large model workflows. Nevertheless, most previous approaches adopt static pipelines, lacking flexibility to generalize to shifting scenarios. Our approach further integrates systematic prompting for reasoning and step-wise tool invocation to address multi-modal tasks.

**Knowledge-based Visual Question Answering.** Knowledge-based Visual Question Answering requires external or specialized knowledge beyond the images themselves to answer questions. Early datasets such as OK-VQA (Marino et al., 2019), KVQA (Shah et al., 2019) and A-OKVQA (Schwenk et al., 2022) focus on general or commonsense knowledge, while more recent benchmarks like ScienceQA (Saikh et al., 2022), Encyclopedic-VQA (Mensink et al., 2023) and InfoSeek (Chen et al., 2023c) introduce more challenging, entity-centric questions requiring intensive knowledge. To solve knowledge-based VQA tasks, some methods fine-tune VLMs to retrieve knowledge from certain knowledge bases. Wiki-LLaVA (Caffagni et al., 2024) employs hierarchical retrieval to augment VLMs with relevant passages from a multi-modal knowledge base, while ReflectiVA (Cocchi et al., 2024) and MR$^2$AG (Zhang et al., 2024) leverage self-reflection mechanisms to reflect on retrieved knowledge relevance and give better retrieved results. Unlike RAG-based methods with fixed pipelines and limited reflection for reranking retrieved knowledge, our agent flexibly coordinates multiple tools with step-by-step reasoning, allowing reflection to guide the entire reasoning and tool selection process for greater effectiveness.

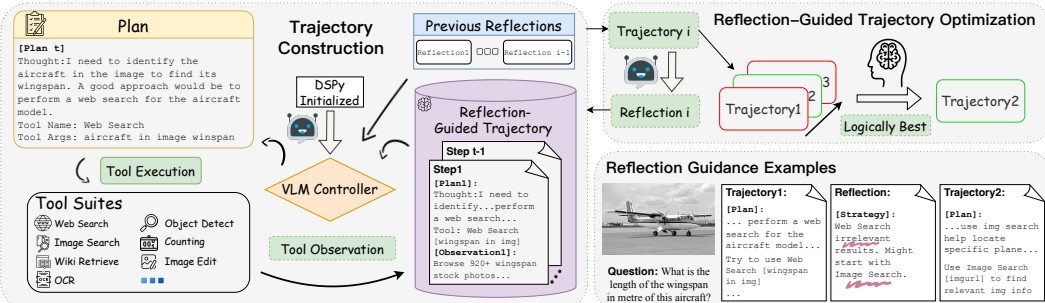

Figure 1: Overall pipeline of our Cross-Reflect framework. The process begins with trajectory construction (left), where the VLM controller is initialized once through DSPy optimization and, at each step $t$, generates a new plan $Plan_t$ based on the question, image, previous trajectory steps, and accumulated previous reflections. Tool execution produces observations that are recorded into the reflection-guided trajectory, which grows step by step. Reflection-guided trajectory optimization (right) then produces a structured reflection $Reflection_i$ with the same lightweight VLM after each complete $Trajectory_i$, conditioning the rollout of $Trajectory_{i+1}$. After all trajectories are generated, the same VLM or a stronger model performs one-time cross-trajectory selection to determine the final trajectory. The bottom right illustrates examples of reflection guidance used to improve subsequent rollouts.

**LLM and VLM Agent** Programming large models can also be viewed as constructing agents. Existing literature has explored using LLMs (Yao et al., 2023; Chen et al., 2023a; Qiao et al., 2024; Yang et al., 2023; Castrejon et al., 2024) and VLMs (Gao et al., 2024; Cheng et al., 2025) as controllers for various tasks, such as reasoning, web search and embodied tasks. Compared to LLM-based agents, VLM-based agents better integrate visual and textual information for visual reasoning tasks. Recently, a concurrent work, OmniSearch (Li et al., 2024b), proposes to dynamically adapt the use of tools in VLM-based agents according to the external knowledge. In contrast, our approach not only employs a VLM as the agent controller for effective visual-text reasoning but also enhances performance through test-time scaling with reflection, allowing for more robust multi-step reasoning in complex visual environments.

**Reflection mechanism.** Reflection, broadly defined as a model's ability to review and revise its own outputs, has been shown to improve performance in LLM, VLM and even generation domains (Shinn et al., 2023; Lei et al., 2024; Zhuo et al., 2025; Team et al., 2025). Self-reflection has been widely used in LLMs, which helps models to identify errors and iteratively refine answers, leading to better reasoning and robustness (Shinn et al., 2023; Madaan et al., 2023). Recent work extends these ideas to VLMs, where reflection helps models detect mistakes in visual reasoning and adjust their responses accordingly (Gao et al., 2024). Despite the advantages, most existing approaches construct datasets and fine-tune models to achieve self-reflection (Yuan et al., 2025; Xu et al., 2025). We focus on a structured prompting framework that incorporates both guided reflection and action-thinking dual reflection to enable self-reflection and self-improvement.

## 3 METHOD

We introduce Cross-Reflect, an inference-time framework that equips multimodal agents with trajectory-level reflection and optimization. Instead of binding reasoning to handcrafted prompt templates or committing to a single sequence of thoughts and actions, Cross-Reflect treats each rollout as a complete trajectory that can be evaluated as a whole. After execution, the trajectory is analyzed for logical consistency, tool ordering, and overall effectiveness. Structured reflections from this analysis are then incorporated into subsequent rollouts, allowing the agent to progressively refine its reasoning. A final cross-trajectory selection step identifies the most coherent solution. By shifting from step-level supervision to trajectory-level reflection, Cross-Reflect provides a simple yet effective mechanism to enhance reliability without reward models, fine-tuning, or additional data.

## 3.1 Modular Multimodal Agent Programming

We build Cross-Reflect upon a modular multimodal programming abstraction that extends the DSPy framework to vision-language tasks. Based on DSPy (Khattab et al., 2024), our framework requires signatures to define the inputs and outputs, while the optimization of prompting and its interaction with VLMs are handled internally in a modularized manner. A signature in DSPy is a declarative specification of a transformation function (e.g., `"question → answer"`), which abstracts what the model should accomplish rather than how it should be prompted. To be more specific, the signature is defined as `"question, image → answer"`. As the original DSPy primarily focuses on language-only settings, we extend it by integrating image modality alongside textual information, ensuring that multimodal inputs are consistently preserved throughout the process.

As illustrated in Fig. 1, the agent's reasoning starts with an iterative planning loop managed by a lightweight VLM controller. At each step, the controller formulates a plan specifying the next reasoning intention, the tool to invoke, and the corresponding arguments. The chosen tool is then executed, producing an observation from either text or image input. This observation is integrated into the agent's memory bank, updating the global trajectory. The trajectory thus serves as a structured record of plans and observations, which later provides the basis for reflection and optimization. The process terminates once the controller indicates that sufficient information has been gathered, and the accumulated trajectory is then used to derive the final answer.

As a benefit of this modular integration, the well-developed prompting modules in DSPy can be easily adapted to our setting. We employ the BootstrapFewShot optimizer (Khattab et al., 2024) to automatically select in-context examples for agent initialization. This optimization is performed once using a held-out subset of data, after which the selected examples are stored and reused for subsequent calls. With minimal manual effort, we obtain a pre-optimized baseline pipeline that already provides strong reasoning capability, and which can be further improved through the reflection-guided optimization introduced in the next section.

---

**Prompt: Structured Reflection Prompt**

**System Prompt:**
You are an expert in analyzing and improving the reasoning and tool usage strategies of agents solving visual question answering (VQA) tasks and provide concise suggestions. You have access to the following tools: {Tools and tool descriptions}.

**User Prompt:**
Given the following VQA question {Question, Image} and the trajectory generated by an agent {Tool Invocation Strategies}, evaluate if the tool usage sequence was logical, effective, and diverse, avoiding redundancy. {Guidance for each aspect}.
Assess if each step in the trajectory was accurate and had no flaws impacting the trajectory.
Output a concise summary suggestion for each aspect. If everything is optimal, output None.

---

Figure 2: Structured reflection prompt used in Cross-Reflect for generating feedback over trajectories.

## 3.2 Reflection-Guided Trajectory Optimization

While the modular programming introduced in Section 3.1 enables lightweight VLM controllers to plan and execute tool calls, their reasoning often suffers from suboptimal tool ordering, redundant invocations, or logical inconsistency. These errors arise naturally from the limited reasoning capacity of small VLMs and can accumulate across steps, ultimately leading to the incorrect final answer. To address this issue, we design a reflection-guided trajectory optimization framework that allows the agent to refine its reasoning holistically and improve reliability without external supervision.

**Reflection Guidance.** After each trajectory is generated, we employ a collaborative reflection module to provide targeted reflections and steer subsequent agent behavior. As shown in Fig. 2, once receiving a trajectory, this module uses a structured prompt to instruct the agent to critically analyze its trajectory along several axes: the tool invocation strategy (including order, utility, and diversity of tool calls) and the step-by-step logical coherence of its reasoning process.

A key aspect of our reflection design is its explicit focus on tool usage order. For example, we observe in the experiment that we can improve the performance by manually starting the plan with a Google Lens Image Search tool. Thus, we encourage reflection not only on the correctness of a specific tool but also on the global order and diversity of the tools. The reflection process also asks the agent to identify redundant or missing tool calls and to comment on whether all relevant information was extracted at each step.

This reflection feedback is formulated through structured prompts and injected into the subsequent rollout, thereby making the agent conditioned on prior critiques. Compared to random sampling, which increases trajectory diversity without offering guidance, reflection explicitly directs future reasoning toward improved strategies. In contrast to reward-model-based methods that require costly training and task-specific data, reflection serves as a lightweight and interpretable supervisory signal that can be applied universally in a training-free setting. Specifically, we incorporate reflections by modifying the system prompt at the start of each new sampling iteration. We prepend the following instruction to the prompt:

---

**Prompt: Reflection Feedback**

You have attempted to answer the following question before. {Question, Image}. If previous reflections are None, please ignore them. If previous reflections are not None, please use them to improve your strategy for correctly answering the given question. {Previous reflections},

---

Figure 3: Reflection feedback prompt.

In our experiments, we also evaluated the setting with more advanced models to provide reflection feedback. However, we found that self-reflection, using the same model as the controller, better aligns with the agent's own reasoning style and yields more practical guidance for improvement.

**Trajectory Selection.**   Once a pool of candidate trajectories has been generated and refined, Cross-Reflect performs a final selection step to identify the most coherent and well-supported reasoning path. In this stage, only trajectories are provided to a model that judges logical soundness and completeness. The selection can be carried out by the same lightweight controller for efficiency, or by a stronger VLM controller for higher accuracy. Since this decision is made only once regardless of the number of candidates, the additional computation is negligible relative to trajectory generation. The chosen trajectory then yields the final answer, ensuring that the output reflects both diverse exploration and reflection-driven optimization.

Overall, our reflection–refinement–selection pipeline elevates reflection from a step-level correction to a trajectory-level optimization principle. By combining structured reflections with cross-trajectory evaluation, Cross-Reflect achieves consistent improvements in robustness and reliability without relying on fine-tuning or reward models.

### 3.3 IMPLEMENTATION WITH TOOL SUITES

To instantiate Cross-Reflect in practice, we integrate an example suite of external tools through a unified interface. The tool layer covers three main categories: retrieval (web search, image search, and Wikipedia page retrieval), perception (OCR and object detection), and counting (estimating object multiplicities from detected regions). Each tool adheres to a consistent input–output signature so that its outputs can be directly incorporated into the trajectory memory. This design emphasizes categories rather than specific providers: any API that conforms to the same contract can be substituted without altering the framework. Detailed descriptions of tool providers, implementation strategies, and integration settings can be referred to Appendix A.

### 4 EXPERIMENTS

#### 4.1 EXPERIMENTAL SETUP

**Datasets.**   Our experiments are conducted on three knowledge-intensive VQA benchmarks: InfoSeek (Chen et al., 2023c), Encyclopedic VQA (EncVQA) (Mensink et al., 2023), and Dyn-VQA (Li et al., 2024b).

Table 1: Main results on knowledge-based VQA, compared with advanced training-free methods. **Bold number** represents the best performance and underline number represents the second best performance.

| Method | INFOSEEK_Wikidata | | | INFOSEEK_Human | | | INFOSEEK_Validation | | | EncVQA | | | Dyn-VQA | Avg |
|---|---|---|---|---|---|---|---|---|---|---|---|---|---|---|
| | Unseen Question | Unseen Entity | All | Unseen Question | Unseen Entity | All | Unseen Question | Unseen Entity | All | Single Hop | Two Hop | All | en | |
| *Proprietary Model* | | | | | | | | | | | | | | |
| Zero-Shot | 20.6 | 18.0 | 19.3 | 18.4 | 10.6 | 14.5 | 23.6 | 18.8 | 21.2 | 32.2 | 35.0 | 33.6 | 30.8 | 23.9 |
| ICL (Brown et al., 2020) | 25.4 | 21.4 | 23.4 | 20.0 | 15.4 | 17.7 | 28.2 | 22.8 | 25.5 | 35.8 | 38.8 | 37.3 | 27.1 | 26.2 |
| ReAct (Yao et al., 2023) | 36.8 | 37.2 | 37.0 | 29.0 | 20.6 | 24.8 | 39.4 | 38.2 | 38.8 | 41.0 | 38.6 | 39.8 | 50.3 | 38.1 |
| ReAct+ (Burns et al., 2023) | 37.8 | 38.0 | 37.9 | 30.6 | 20.4 | 25.5 | 40.8 | 38.6 | 39.7 | 46.0 | **42.8** | 44.4 | 51.9 | 39.9 |
| ReAct-BoN (Snell et al., 2024) | 38.0 | 39.2 | 38.6 | 31.8 | **24.0** | 27.9 | 44.0 | 36.6 | 40.3 | 37.6 | 36.0 | 36.8 | 51.9 | 39.1 |
| HAMMR (Castrejon et al., 2024) | – | – | – | – | – | – | – | – | – | 47.8 | 22.0 | 34.9 | | – |
| OmniSearch (Li et al., 2024b) | – | – | – | – | – | – | – | – | – | – | – | – | 53.2 | – |
| Cross-Reflect | 41.5 | 40.3 | 40.9 | 33.2 | 21.8 | 27.5 | 44.4 | 42.8 | 43.6 | **51.6** | **40.2** | **45.9** | 52.8 | 42.1 |
| Cross-Reflect + | **42.6** | **44.6** | **43.6** | **34.0** | 23.2 | **28.6** | **46.2** | **44.8** | **45.5** | 51.5 | 39.2 | 45.4 | **54.1** | **43.4** |
| *Open-Source Model* | | | | | | | | | | | | | | |
| Zero-Shot | 15.2 | 16.4 | 15.8 | 14.0 | 11.6 | 12.8 | 19.6 | 18.8 | 19.2 | 16.0 | 13.2 | 14.6 | 27.4 | 18.0 |
| ICL (Brown et al., 2020) | 17.4 | 19.2 | 18.3 | 15.8 | 13.2 | 14.5 | 21.2 | 19.8 | 20.5 | 31.9 | 24.8 | 28.4 | 28.4 | 22.0 |
| ReAct (Yao et al., 2023) | 20.0 | 20.8 | 20.4 | 20.8 | 20.6 | 20.7 | 20.6 | 20.2 | 20.4 | 28.2 | 24.6 | 26.4 | 38.4 | 25.3 |
| ReAct+ (Burns et al., 2023) | 28.6 | 32.4 | 30.5 | 22.2 | 14.8 | 18.5 | 33.2 | 29.6 | 31.4 | 38.5 | **37.1** | **37.8** | 43.6 | 32.4 |
| ReAct-BoN (Snell et al., 2024) | 22.3 | 24.2 | 23.3 | 19.4 | 14.8 | 17.1 | 31.6 | 26.4 | 29.0 | 37.5 | 27.6 | 32.6 | 36.1 | 27.6 |
| OmniSearch (Li et al., 2024b) | – | – | – | – | – | – | – | – | – | – | – | – | 43.2 | – |
| Cross-Reflect | 31.2 | 34.6 | 32.9 | 28.2 | 20.4 | 24.3 | 33.0 | 28.2 | 30.6 | 41.7 | 26.0 | 33.9 | 40.3 | 32.4 |
| Cross-Reflect + | **35.0** | **37.4** | **36.2** | **32.2** | **21.2** | **26.7** | **40.0** | **36.0** | **37.0** | **48.2** | 32.6 | 36.4 | **46.1** | **36.5** |

InfoSeek consists of 1.3M image-question-answer triplets aligned with approximately 11K Wikipedia entities. The dataset offers three evaluation subsets: INFOSEEK_Wikidata, INFOSEEK_Human, and INFOSEEK_Validation. The INFOSEEK_Wikidata and INFOSEEK_Validation subsets are automatically constructed from Wikipedia and Wikidata, while INFOSEEK_Human contains 8.9K human-annotated samples designed to better simulate real-world information-seeking scenarios. Each subset is further divided into unseen entity and unseen question splits to test model's generalization.

EncVQA comprises 221K question-answer pairs associated with 16.7K fine-grained entities, with up to five images per entity. The dataset includes single-hop and two-hop subsets: single-hop questions require reasoning over a single Wikipedia article, while two-hop questions require linking two pieces of information from the Wikipedia-based knowledge base, following a rule-based collection approach. Each question-answer pair is linked to the relevant Wikipedia articles and supporting evidence paragraphs, with all samples drawn from a controlled knowledge base of 2M Wikipedia pages.

Dyn-VQA is a recently introduced benchmark targeting dynamic knowledge retrieval for visual question answering. It contains questions that explicitly depend on time-sensitive information, requiring access to dynamic knowledge sources rather than static encyclopedic knowledge. The benchmark includes both entity-based and event-based queries, making it a challenging benchmark for evaluating whether agents can adapt retrieval strategies to handle non-stationary knowledge.

**Evaluation.** We follow the standard evaluation protocols provided by each dataset. For InfoSeek, evaluation metrics vary according to the question type. Specifically, questions in the STRING and TIME categories are evaluated using standard VQA Accuracy (Goyal et al., 2017), while NUMERICAL questions are assessed using Relaxed Accuracy (Methani et al., 2020). For EncVQA, we employ the BERT Matching Score (BEM) (Bulian et al., 2022) following the official evaluation protocol, which measures semantic similarity between predicted and ground-truth answers using BERT-based embeddings. For Dyn-VQA, we report exact match accuracy according to the benchmark.

**Implementation Details.** Our generation pipeline is based on the DSPy (Khattab et al., 2024) library. For the controller model, we consider both proprietary models, `GPT-4o-mini` (Hurst et al., 2024), and open-source VLMs, `Qwen2.5-VL-7B` (Bai et al., 2025). The Qwen2.5-VL-7B model is deployed using the vLLM (Kwon et al., 2023) framework. All experiments are conducted with the official inference hyperparameters.

In Cross-Reflect +, plan and reflection generation is carried out by the same controller model as Cross-Reflect, while trajectory selection is handled by a single call to GPT-4o.

**Baselines.** We designed and implemented the following works for a fair comparison in our multimodal setting.

- **Zero-Shot**. The models are prompted to generate VQA answers without any additional retrieval or reasoning steps. We implement it with the `predict` module in DSPy.
- **ICL** (Brown et al., 2020). The models employ `BootstrapFewShot` optimizer to optimize the in-context examples. Then, at each time sampling, the model is conditioned on the collected examples to give the final answer. We also test on a more advanced MiPro optimizer (Khattab et al., 2024), but find that it actually costs much more to optimize, and the performance is only slightly improved; see the supplementary material for more details.
- **ReAct** (Yao et al., 2023). The models generate an explicit reasoning trajectory in the form of alternating `[Thought]-[Tool]-[Observation]` steps.
- **ReAct+**. Following the weak-to-strong paradigm (Burns et al., 2023), the models employ ReAct for plan generation using our smaller model, then utilize a stronger model (`GPT-4o`) to produce the final answer.
- **ReAct-BoN** (Snell et al., 2024). The models employ ReAct for plans and Best-of-N for test-time scaling. We apply the same stronger model as Cross-Reflect + to select the answer. In practice, we choose N=3 for our experiments.
- **HAMMR** (Castrejon et al., 2024). This model introduces a hierarchical multimodal ReAct system where agents can invoke specialized sub-agents, enhancing compositionality and improving the accuracy of LLM+tools approaches across diverse VQA tasks.
- **OmniSearch** (Li et al., 2024b). This model introduces a retrieval-augmented multimodal agent that unifies text, image, and web search, and employs dynamic query decomposition with iterative retrieval to handle time-sensitive knowledge queries.

Throughout the baselines, we normalize the cost of the model calling, that is, we allow at most one call to the stronger model.

## 4.2 MAIN RESULTS.

Table 1 reports results across InfoSeek, EncVQA, and Dyn-VQA. Zero-Shot and ICL baselines confirm the limited capacity of current VLMs for knowledge-intensive VQA, with INFOSEEK$_{\text{Wikidata}}$ scores remaining below 25. Retrieval-augmented agents such as ReAct substantially improve over these simple baselines, but their gains remain moderate; variants like ReAct+ and ReAct-BoN provide only incremental improvements, suggesting that test-time sampling or stronger answer synthesis alone is insufficient.

By contrast, **Cross-Reflect** consistently achieves the strongest performance among training-free approaches. On InfoSeek$_{\text{Wikidata}}$, it improves the Overall score by more than three points over ReAct with GPT-4o-mini, and by over twelve points with Qwen2.5-VL-7B. On EncVQA single-hop, it reaches 51.6 with GPT-4o-mini, the highest among all prompt-based methods. Allowing one stronger model call for selection (**Cross-Reflect +**) yields further gains, pushing InfoSeek$_{\text{Wikidata}}$ to 44.6 and EncVQA single-hop to 51.5, confirming the effectiveness of reflection-guided sampling with large-model selection.

A similar trend is observed on Dyn-VQA, which emphasizes dynamic knowledge retrieval. OmniSearch (Li et al., 2024b), a fine-tuned retrieval-augmented agent, achieves 43.23 with Qwen-VL-Chat and 53.21 with GPT-4o. Remarkably, **Cross-Reflect +** (Qwen2.5-VL-7B) attains 46.13 without any fine-tuning, surpassing OmniSearch despite relying only on a training-free controller. When controlled by GPT-4o-mini and make only one GPT-4o call for selection, **Cross-Reflect +** further improves to 54.14, again outperforming OmniSearch(GPT-4o).

## 4.3 COMPARISON TO TRAINING-BASED METHODS

Table 2 provides a comparison between our training-free framework and a diverse set of state-of-the-art fine-tuned methods. Wiki-LLaVA (Caffagni et al., 2024) builds on the LLaVA (Liu et al., 2024) model and trains with large-scale multimodal datasets of approximately one million samples, incorporating a hierarchical retrieval pipeline that demands substantial computational resources. EchoSight (Yan & Xie, 2024) leverages a pretrained LLaMA3 initialized with CLIP and fine-tunes it on encyclopedic VQA trainsets with around five hundred thousand samples, requiring significant retraining costs for its reranking-based retrieval mechanism. ReflectiVA (Cocchi et al., 2024) starts from LLaMA-3.1 and employs a two-stage training process on task-specific datasets with about fifty

Table 2: Comparison of our method with training-based methods. **Bold number** represents the best performance and underline number represents the second best performance.

| Method | INFOSEEK$_{Wikidata}$ | | | INFOSEEK$_{Human}$ | | | INFOSEEK$_{Validation}$ | | | EncVQA | | |
|---|---|---|---|---|---|---|---|---|---|---|---|---|
| | Unseen Question | Unseen Entity | Overall | Unseen Question | Unseen Entity | Overall | Unseen Question | Unseen Entity | Overall | Single Hop | 2 Hop | Overall |
| CLIP -> PALM (Chen et al., 2023c) | 21.9 | 18.6 | 20.1 | 15.6 | 14.9 | 15.2 | 22.7 | 18.5 | 20.4 | – | – | – |
| CLIP -> FID (Chen et al., 2023c) | 20.7 | 18.1 | 19.3 | 18.9 | 17.6 | 18.2 | 23.3 | 19.1 | 20.9 | – | – | – |
| Wiki-LLAVA (Caffagni et al., 2024) | – | – | – | – | – | – | 30.1 | 27.8 | 28.9 | 17.7 | – | – |
| EchoSight (Yan & Xie, 2024) | – | – | – | – | – | – | – | – | 31.3 | 41.8 | – | – |
| ReflectiVA (Cocchi et al., 2024) | – | – | – | – | – | – | 40.4 | 39.8 | 40.1 | 28.0 | – | – |
| mR$^2$AG (Zhang et al., 2024) | 39.1 | 38.0 | 38.6 | 30.2 | **27.5** | **28.8** | 40.6 | 39.8 | 40.2 | **55.9** | – | – |
| Cross-Reflect | 41.5 | 40.3 | 40.9 | 33.2 | 21.8 | 27.5 | 44.4 | 42.8 | 43.6 | 51.6 | **40.2** | **45.9** |
| Cross-Reflect + | **42.6** | **44.6** | **43.6** | **34** | 23.2 | 28.6 | **46.2** | **44.8** | **45.5** | 51.5 | 39.2 | 45.4 |

Table 3: Ablations on tool order influence.

| | Unseen Question | Unseen Entity | Overall |
|---|---|---|---|
| ReAct | 39.4 | 38.2 | 38.8 |
| + Fixed | 41.4 | 37.8 | 39.6 |
| + Uniform | 43.0 | 41.2 | 42.1 |

Table 4: Ablations on effectiveness of reflection guidance.

| | Unseen Question | Unseen Entity | Overall |
|---|---|---|---|
| ReAct | 39.4 | 38.2 | 38.8 |
| + Uniform | 43.0 | 41.2 | 42.1 |
| + Reflection | 44.4 | 42.8 | 43.6 |

Table 5: Ablations on choice of reflection model.

| | Unseen Question | Unseen Entity | Overall |
|---|---|---|---|
| o3-mini | 43.8 | 41.2 | 42.5 |
| 4o-mini | 46.2 | 44.8 | 45.5 |

thousand samples, which requires moderate computational resources. mR$^2$AG (Zhang et al., 2024) is based on the LLaVA model and fine-tuned on the mR²AG-IT dataset containing approximately one hundred thousand samples, with additional overhead introduced by its retrieval-reflection mechanism. Remarkably, despite not relying on any task-specific fine-tuning, our approach achieves highly competitive, and in some cases superior, results. On INFOSEEK$_{Wikidata}$, Cross-Reflect and Cross-Reflect + achieve Overall scores of 40.9 and 43.6, respectively, outperforming most fine-tuned baselines such as mR$^2$AG, which achieves 38.6, and CLIP→PALM, which achieves 20.1. Strong results are also observed on INFOSEEK$_{Validation}$ and EncVQA.

Overall, these results demonstrate the strong generalization and efficiency of our training-free framework. Without any task-specific fine-tuning, our method not only surpasses prompt-based agent baselines but also matches several state-of-the-art fine-tuned approaches on challenging knowledge-intensive VQA benchmarks. This highlights the practical value of our approach, especially for real-world scenarios where large-scale annotation or model retraining is impractical.

## 4.4 Ablation Studies

In this section, we conduct ablation studies to verify the effectiveness of different components in our framework, including the tool selection strategy, the impact of reflection, and the choice of reasoning model for reflection on the **INFOSEEK$_{Validation}$** dataset.

**Effect of Tool Order** We observe that the order of the tools is important to the final performance. As such, we evaluate static strategies of tool plan compared to our reflection-based tool plan in Table 3. We first conduct the ReAct+Fixed baseline, where we fix the first tool to a specific tool, such as Google Lens Image Search. This significantly improves the model performance of the naïve ReAct baseline, suggesting that a strong initial tool choice can benefit the reasoning trajectory. To further analyze the impact of tool order, we design a "Uniform" strategy: the model first generates its top-3 relevant tools, then we sample three runs, each time using one of these tools as the first tool, and finally select the best answer. We find that covering more potential tool invocation orders leads to substantial gains. On INFOSEEK$_{Validation}$, the unseen entity score improves from 37.8 to 41.2, and the unseen question score from 41.4 to 43.0. This demonstrates the importance of a more flexible and advanced tool selection strategy. Nonetheless, these static strategies still require multiple inference runs and lack adaptability. In contrast, our full method employs a dynamic, reflection-based approach to adaptively determine optimal tool orders.

**Effect of Reflection.** We next ablate the impact of the reflection mechanism. In the uniform sampling setting above, the only guidance provided is for the first tool selection; in contrast, our full reflection approach provides guidance not only for tool choice but also for each step of the

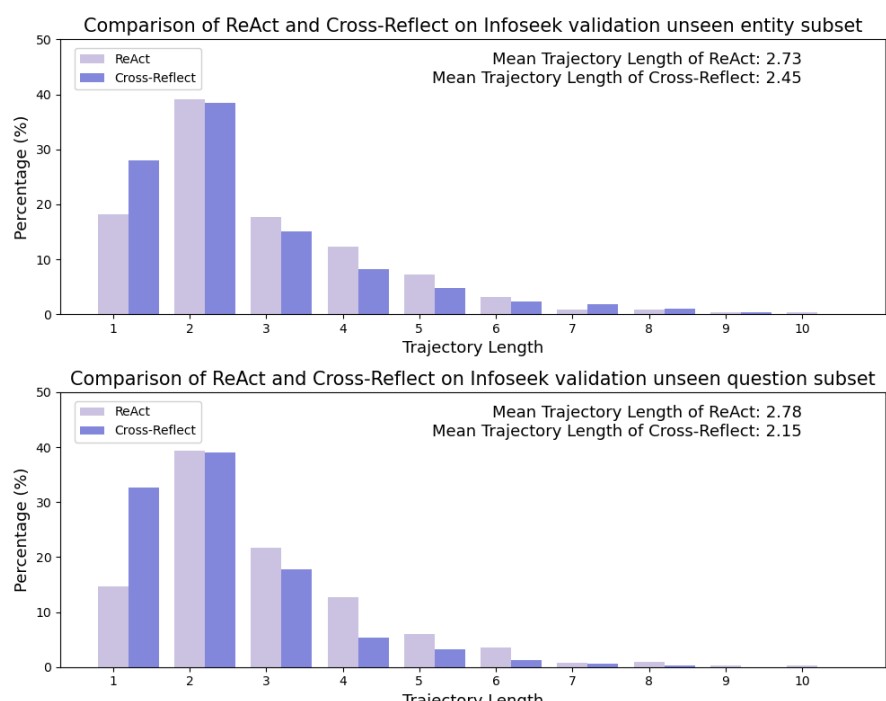

Figure 4: Distribution of trajectory lengths for Qwen2.5-VL-7B on the Infoseek validation set.

reasoning process. As shown in Table 4, adding reflection leads to consistent improvement over uniform sampling. For instance, on INFOSEEK$_{\text{Validation}}$, the unseen entity accuracy rises from 41.2 to 44.8, and unseen question from 43.0 to 46.2. This highlights the benefit of structured, trajectory-level reflections for improving complex multi-hop reasoning.

**Effect of Reflection Models.** Finally, we study the effect of using different models for providing reflections. With the recent surge of strong reasoning models (Guo et al., 2025), we explore replacing our default reflection model (GPT-4o-mini) with o3-mini. Interestingly, as shown in Table 5, using o3-mini for reflection yields lower performance. On INFOSEEK$_{\text{Validation}}$, the unseen entity score improves from 41.2 to 44.8, and the unseen question score from 43.8 to 46.2. We hypothesize that our task does not require very complex or abstract reasoning, and that GPT-4o-mini already provides sufficiently effective guidance for this setting.

**Effect on Trajectory Efficiency.** We further analyze whether Cross-Reflect improves efficiency by shortening reasoning trajectories. We define trajectory length as the number of `Plan-Execution-Integration` steps within a trajectory. For Qwen2.5-VL-7B, Cross-Reflect substantially reduces the average trajectory length from 2.73 to 2.45 on the unseen entity split, and from 2.78 to 2.15 on the unseen question split. As shown in Fig. 4, the distribution shifts toward shorter paths, with a clear increase in single-step trajectories and a reduction in longer rollouts. This indicates that reflection not only improves accuracy but also guides agents toward more concise and efficient reasoning.

## 5 CONCLUSION

In this paper, we propose Cross-Reflect, a modular multimodal agent framework that extends the DSPy paradigm to vision-language reasoning. By introducing reflection-guided trajectory optimization, our method generates multiple candidate trajectories, produces structured reflections after each trajectory to guide subsequent rollouts, and finally performs a one-time cross-trajectory selection to identify the most reliable reasoning path. Extensive experiments demonstrate that Cross-Reflect achieves state-of-the-art results among training-free methods and competitive performance compared to fine-tuned baselines, highlighting its robustness and adaptability. We believe this work provides a scalable foundation for building self-reflective multimodal agents and opens promising directions for future progress in open-world visual reasoning.

ETHICS STATEMENT

Our work builds modular multimodal agents that leverage reflection-guided optimization for knowledge-intensive VQA. We do not collect or annotate any new human subject data; all experiments are conducted on publicly available datasets (InfoSeek, EncVQA, and Dyn-VQA) released under research licenses. We follow the original dataset creators' terms of use and cite them appropriately. As our approach operates purely at inference time without fine-tuning, it does not introduce additional risks of data misuse or privacy leakage. Potential risks include the propagation of biases present in the underlying datasets or vision-language models; we acknowledge this limitation and emphasize that our method does not mitigate dataset-level biases but can improve reasoning robustness. No personally identifiable information or sensitive content was involved in this study.

REPRODUCIBILITY STATEMENT

We make every effort to ensure reproducibility of our results. All implementation details of Cross-Reflect are described in Section 4, including model initialization through DSPy, reflection-guided trajectory optimization, and one-time cross-trajectory selection. The datasets used in our experiments are publicly available: InfoSeek (Chen et al., 2023c), EncVQA (Mensink et al., 2023), and Dyn-VQA (Li et al., 2024b). We report all evaluation protocols, and metrics in Section 4, and provide ablation studies to analyze the effect of key components. Our prompts are also provided in Section 3 and Appendix.

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

## A    HYPERPARAMETERS AND EXPERIMENTAL DETAILS

**Prompts**    The main prompts in our experiments are formatted automatically by DSPy, which generates input prompts for each task and model according to its internal template logic. As a result, most of the prompt structure is determined by DSPy itself.

The only component we want to explain more specifically is the tool description, which provides textual information about each tool available to the model. Below we provide examples of the tool names and their corresponding description as used in our experiments:

---

### Tool Descriptions

**Tool Name:** `Web Search`
**Description:** A wrapper around Google Search API. Useful for when you need extra textual information about the given query. It cannot process image input, but only textual input. If it requires extra information about an image, do Google Lens Image Search first. Input should be a text string.

**Tool Name:** `Google Lens Image Search`
**Description:** A wrapper around the Google Lens Image Search API. This tool can use the image URL to search similar images across the web. It can find relevant image titles and sources and return them as a text string. Input should be an image URL.

**Tool Name:** `OCR Understanding`
**Description:** A wrapper around OCR Understanding (Optical Character Recognition). Useful when you find that text or handwriting is crucial to answer the question. This tool can find the actual text, written name, or product name in the image. Input should be an image URL, or a PIL Image.

**Tool Name:** `Wikipedia Page`
**Description:** A wrapper around Wikipedia Page retrieval. Useful for retrieving full Wikipedia page contents based on a given topic or query. Input should strictly be a concise keyword (e.g., a name or topic) rather than a full question or a keyword plus a part question extended phrase. The tool will return relevant content from Wikipedia based on the provided keyword.

**Tool Name:** `Detect Object`
**Description:** A wrapper around object detection functionality. Useful for identifying and locating objects within an image, along with their categories and bounding boxes. Input should be an image (URL or PIL Image), and the tool will return detected objects with their details.

**Tool Name:** `Image Edit`
**Description:** A wrapper around image manipulation functionality. Useful when you find a specific area of the image is important. Should be used after the Detect Object tool. If the Detect Object tool detected useful objects, it would draw a red bounding box with the Detect Object tool results. If not, it would crop the image with a 200*200 bounding box, according to the args. Input should include bounding box coordinates (x1, y1, x2, y2).

**Tool Name:** `Count Object`
**Description:** A wrapper around object counting functionality. Useful for counting the number of objects within an image. Should be used after the Detect Object tool. Input should include target object bounding box coordinates (x1, y1, x2, y2), and the tool will return the number of objects detected.

---

**Experimental Environment**    All experiments were conducted using DSPy version 2.5.15. For model inference, GPT-4o-mini was run without GPU acceleration, while Qwen2.5-VL-7B was evaluated using a single NVIDIA A100 GPU.

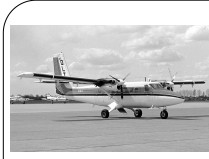

**Trajectory1:**
**[Plan1]:**
Thought: I need to identify the aircraft in the image to find its wingspan. A good approach would be to perform a web search for the aircraft model.
Tool: Web Search[aircraft in image wingspan]
**[Observation1]:** Useless information. Lead to failure.
**[Finish]:** 17.68 meters

**Question:** What is the length of the wingspan in metre of this aircraft?

**Reflection:**
**[Tool Invocation Strategy]:** The sequence was logical, but starting with Google Lens Image Search instead of a web search could have directly identified the aircraft, streamlining the process. The web search yielded only partially relevant results, making it somewhat unnecessary before identification. The agent effectively used the Wikipedia tool after identification.

**[Step-wise Thinking]:** The agent's reasoning was mostly accurate; however, the initial web search step did not contribute directly to identifying the aircraft model and could indicate a misunderstanding of how to efficiently gather specific information on wingspan.

**Trajectory2:**
**[Plan1]:**
Thought: I need to identify the aircraft in the image to find its wingspan. Using Google Lens Image Search will help me find relevant information about this specific aircraft model.
Tool: Google Lens Image Search[image]
**[Observation1]:** DHC-6_253

**[Plan2]:**
Thought: The aircraft appears to be a DHC-6 Twin Otter based on the search results. I will now look up the wingspan of the DHC-6 Twin Otter on Wikipedia to get the specific measurement
Tool: Wikipedia Page[DHC-6 Twin Otter]
**[Observation2]:** 65 ft 0 in (19.81 m) span
**[Finish]:** 19.8 meters

**Trajectory1:**
**[Plan1]:**
Thought: The image appears to depict a historical building, likely a church or cathedral. To confirm its purpose, I will perform a search for more information about the structure.
Tool: Wikipedia Page[church]
**[Observation1]:** A Large amount of general church information
**[Finish]:** Catholic church

**Question:** What is this facility dedicated to?

**Reflection:**
**[Tool Invocation Strategy]:** The sequence was logical, but starting with Google Lens Image Search instead of a web search could have directly identified the aircraft, streamlining the process. The web search yielded only partially relevant results, making it somewhat unnecessary before identification. The agent effectively used the Wikipedia tool after identification.

**[Step-wise Thinking]:** The reasoning that the image represents a church was logical, but the transition to searching Wikipedia needed specificity to directly address the query regarding the facility's purpose. The agent should have focused more on the image itself or a relevant descriptive search instead.

**Trajectory2:**
**[Plan1]:**
Thought: The image appears to depict a church or a religious facility. I should perform a Google Lens Image Search to more specific information.
Tool: Google Lens Image Search[image]
**[Observation1]:** Title 1: Entrance to the Spanish Colonial revival style Lobero Theatre with red tile roof. Title 2: Santa barbara

**[Plan2]:**
Thought: The Google Lens search did not yield specific information about the facility in the image. However, it appears to be a church, likely a mission based on its architectural style. I should perform a Wikipedia search for more details.
Tool: Wikipedia Page[Santa Barbara Mission]

**[Observation2]:** Description to Santa Barbara Mission
**[Finish]:** Saint Barbara

Figure 5: Qualitative example where Cross-Reflect is able to fix an error in the original trajectory and recover the correct answer.

**Data Sampling** For each subset of Infoseek, we randomly sampled a total of 1,000 examples, with an even split between 500 samples containing unseen questions and 500 samples containing unseen entities. For EncVQA, 1,000 samples were selected for the single-hop setting and 500 samples for the two-hop setting. Our experimental results report the average score across the evaluation datapoints. However, each algorithm is evaluated once per dataset and model due to computational cost. We apologize for the accidental miscommunication in item 7 of the checklist.

**Method-Specific Hyperparameters** For the ReAct method, we set the maximum number of iterations to 10, used a temperature of 0, and set Best-of-N's N as 3. For DSPy's BootstrapFewShot, we set `max_bootstrapped_demos` to 4 and `max_labeled_demos` to 0.

## B QUALITATIVE RESULTS

To further illustrate the effectiveness of our Cross-Reflect method, we present qualitative examples highlighting two key advantages: (1) the ability to fix errors in the original trajectory, and (2) the capability to iteratively refine reasoning and arrive at a shorter, more accurate trajectory.

As shown in Fig. 5, it demonstrates that the initial trajectory leads to an incorrect answer, but the Cross-Reflect approach is able to identify and correct the error through reflection and selection. This shows the potential of Cross-Reflect to recover from reasoning mistakes that would otherwise persist.

As shown in Fig. 6, it showcases a scenario in which repeated reflection enables the agent to reorganize its reasoning process, ultimately discovering a more direct and concise solution. As a result, the final selected trajectory is not only correct but also shorter than the original baseline trajectory.

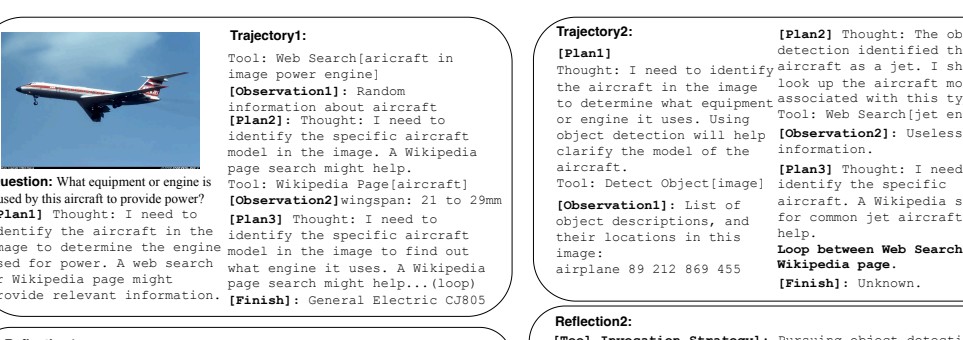

Figure 6: Qualitative example in which Cross-Reflect enables the agent to iteratively refine its reasoning and find a shorter, more direct trajectory to the correct answer.

## C  BROADER IMPACT

Our approach has the potential to benefit a wide range of downstream applications, such as education, healthcare, and information retrieval, by enabling more accurate and efficient reasoning in vision language models. However, the improved performance comes at the cost of increased inference time and computational resources, which may limit its practical deployment in some scenarios. On the other hand, the methods explored in this work are fully compatible with open-source models and can be self-hosted, allowing users to run the system locally and keep all data private. This enables organizations or individuals with stricter privacy requirements to maintain full control over their information, without relying on external servers.

## D  ADDITIONAL ABLATION STUDIES

In this section, we present additional ablation experiments that extend the analyses reported in Section 4.4 of the main paper. These studies provide a more comprehensive view of how different design choices influence the performance of Cross-Reflect and further validate the robustness of our conclusions.

**Effect of Direct Retrieval vs. Reflection.**    To examine whether performance gains come merely from retrieving labels rather than from trajectory-level optimization, we conduct an experiment where the agent is restricted to use only Google Lens Image Search and Wikipedia Page Search. For each image, we apply Google Lens to retrieve the top-10 results and then query Wikipedia using each retrieved title. If Wikipedia returns a relevant page, we provide its content as context; otherwise, we use the image search titles and sources. Importantly, no trajectory generation or reflection-guided optimization is performed—the retrieved text is directly provided to GPT-4o-mini to produce an answer.

As shown in Table 6, direct retrieval yields only 30.2 accuracy, whereas Cross-Reflect achieves 44.4 and 42.8 on unseen question and unseen entity splits, respectively. This large gap demonstrates

Table 6: Comparison of direct retrieval vs. Cross-Reflect on the INFOSEEKValidation set.

| Method | Unseen Question | Unseen Entity |
|---|---|---|
| Direct answer using image search & Wikipedia | 30.2 | 30.2 |
| Cross-Reflect | 44.4 | 42.8 |

that the improvements of Cross-Reflect are not simply due to overlapping retrieval from Wikipedia, but result from the structured reflection-guided trajectory optimization that enables more effective reasoning with retrieved evidence.

**Effect of Large-Model Selection.**    We further analyze the role of large models in our framework by comparing direct zero-shot GPT-4o with our Cross-Reflect pipeline. In Cross-Reflect+, the small model generates candidate trajectories and a single call to GPT-4o is used only for final trajectory selection, without participating in generation or planning. We report results on the INFOSEEKValidation subset in Table 7.

Table 7: Comparison of zero-shot with Cross-Reflect on the INFOSEEKValidation set.

| Method | Unseen Question | Unseen Entity |
|---|---|---|
| Zero-Shot (GPT-4o-mini) | 23.6 | 18.8 |
| Zero-Shot (GPT-4o) | 36.8 | 31.2 |
| Cross-Reflect | 44.4 | 42.8 |
| Cross-Reflect + | 46.2 | 44.8 |

The results show that Cross-Reflect + achieves higher accuracy than using GPT-4o directly in a zero-shot manner, despite calling GPT-4o only once for selection. This demonstrates the efficiency of combining small-model trajectory generation with large-model selection: the approach preserves the cost-effectiveness of lightweight models while leveraging strong models only where they add the most value. Such a hybrid design highlights the practical advantage of reflection-guided trajectory optimization for real-world settings where large-model access is limited or costly.

**Analysis of Token Consumption.**    We further analyze efficiency by reporting output token consumption in addition to trajectory length. This experiment is conducted on 50 randomly selected examples from the INFOSEEKValidation set. Table 8 compares zero-shot prompting, ReAct, ReAct-BoN, and our Cross-Reflect+ pipeline.

Table 8: Comparison of accuracy and output token usage on the Infoseek validation set (50 examples).

| Method | Unseen Question | Unseen Entity | Output Tokens |
|---|---|---|---|
| Zero-Shot | 23.6 | 18.8 | 137 |
| ReAct | 39.4 | 38.2 | 40,993 |
| ReAct-BoN | 44.0 | 36.6 | 116,042 |
| Cross-Reflect + | 46.2 | 44.8 | 135,289 |

The results show that Cross-Reflect + substantially outperforms simpler baselines such as Zero-Shot and ReAct. Moreover, when comparing methods with similar token-level computational budgets, Cross-Reflect + achieves higher accuracy than ReAct-BoN, proving the effectiveness–efficiency trade-off of reflection-guided trajectory optimization.

