# OpenReview forum: "Cross-Reflect: Empowering Multi-Modal Agents with Joint Reasoning Across Trajectories"
_ICLR.cc/2026/Conference — Submitted to ICLR 2026_

### Official Review · Reviewer_7rn3 · 2025-10-29

**Soundness:** 1
**Presentation:** 1
**Contribution:** 2
**Rating:** 2
**Confidence:** 4

**Summary:**

The paper proposes Cross‑Reflect, a training‑free, test‑time framework for VLM agents that (i) samples multiple tool‑use trajectories, (ii) performs structured, trajectory‑level reflection after each rollout, and (iii) conducts a cross‑trajectory selection to pick the final answer. The approach is implemented by extending DSPy from language‑only to multi‑modal agent programming, and integrates a suite of tools (web/image search, Wikipedia, OCR, detection, counting).

**Strengths:**

1. The paper is well motivated, pointing out the need for small VLMs to solve hard VQA problems.
2. The proposed method, Cross-Reflect, shows performance gain compared with training-free QA baselines.

**Weaknesses:**

- The paper is poorly written. The organization of the experimental section (especially Section 4.4) is confusing and lacks a clear logical flow. Figure 1 is poorly presented, making it difficult for readers to understand the content.

- The method is not open-sourced. I strongly urge the authors to release their code to facilitate reproducibility and independent verification.

- The experiments are insufficient and incomplete.

  - The results for HAMMER and OmniSearch are incomplete. The baseline comparison is weak without a thorough evaluation of these recent methods.

  - Cross-Reflect should also be tested on other open-source models. Evaluating only on Qwen-2.5-VL-7B is inadequate, as the prompt design may have been optimized specifically for the model.

  - Section 4.3 (“Comparison with training-based methods”) is unconvincing, since LLaVA is a relatively weak base model compared with Qwen-2.5-VL.

- The paper lacks an in-depth analysis of the proposed method.

  - Does the VLM trajectory improve as more reflections are generated? Does the selector tend to favor later trajectories? How many trajectories are produced per question?

  - The ablation studies are insufficient. What happens if the selector is removed and the final answer is simply taken from the last trajectory? I understand that removing the reflection module makes Cross-Reflect similar to ReAct-BoN, but what if the reflection prompt were incorporated directly into the actor VLM? This could serve as a prior for better problem solving in the “Reflection Guidance” component of the reflector prompt.

- The time cost of the proposed method is unclear. Given the high token usage and computational overhead, the authors should compare it with reasoning-enhanced models of similar size (e.g., Qwen3-VL-8B).

**Questions:**

- Include the prompt used for the selector.

- In Appendix A, the sentence “GPT-4o-mini was run without GPU acceleration” is unprofessional and potentially misleading. Shouldn't it specify that the model was accessed via an API instead?

- I am curious whether the proposed method remains effective in a text-only setting. Are there any components in the design that are specifically tailored for VLMs?

- I also wonder how large a VLM would need to be to achieve comparable performance to Reflect-Cross on 7B-scale models.

---

### Official Review · Reviewer_Lbfm · 2025-10-30

**Soundness:** 3
**Presentation:** 2
**Contribution:** 2
**Rating:** 4
**Confidence:** 3

**Summary:**

This paper explores how small LLMs, which are often considered unreliable due to failures in tool selection and ineffective reasoning trajectory construction, can function as effective agents for coherent multi-step tool use. The motivation lies in their low cost and suitability for large-scale deployment.
Previous methods either focus on reflection limited to final answers or chains of thought, or rely on reward models that require training on curated datasets and therefore lack generality.
Cross-Reflect leverages DSPy with minimal manual prompting to extend to vision-language tasks. It generates multiple candidate trajectories, performs structured reflection to critique and refine them, and conducts cross-trajectory selection to identify the most reliable solution.
Extensive experiments across static and dynamic VQA benchmarks demonstrate that Cross-Reflect consistently improves small VLMs by enabling flexible tool usage and trajectory-level self-reflection, achieving average relative improvements of 11% for proprietary models and 28% for open-source models over baseline methods.

**Strengths:**

* Expands DSPy into the multimodal setting.
* Proposes trajectory-level reflection as an in-context, training-free approach for VQA reasoning.

**Weaknesses:**

* The paper is generally readable but has presentation issues that reduce clarity and make it harder for readers to grasp the main ideas.
   * The description of DSPy in Section 3.1 should appear earlier, for example in the introduction, to help readers establish a shared foundation.
   * Figure 1 and its reference text are too far apart, and the same applies to Table 1.
   * The detailed procedure of cross-trajectory selection is insufficiently explained.

* The set of baselines is limited. It would be valuable to include some search-based algorithms in *Zhu, K., Li, H., Wu, S., Xing, T., Ma, D., Tang, X., Liu, M., Yang, J., Liu, J., Jiang, Y. E., Zhang, C., Lin, C., Wang, J., Zhang, G., & Zhou, W. (2025). Scaling test-time compute for LLM agents. arXiv preprint arXiv:2506.12928*.

* The paper lacks a detailed analysis explaining why Cross-Reflect outperforms the baselines. See the `Questions` section below for specific suggestions.

* Since Cross-Reflect accumulates previous trajectory histories for subsequent planning, it represents a tradeoff between planning within a single trajectory (based on reflection on failed states) and planning across multiple trajectories (based on trajectory-level reflection). While this improves performance, it also increases computational cost. Efficiency should be discussed in more depth. Notice that Table 8 shows that moving from zero-shot to Cross-Reflect+ yields 96% and 138% performance gains, but token consumption increases nearly 1000 times, which may indicate that the method is not very efficient in terms of computational cost.
* The generalizability of Cross-Reflect appears limited. For a new task, even within the same domain, the method requires generating new trajectory histories with corresponding reflections.

**Questions:**

* Why does Cross-Reflect achieve larger performance gains on proprietary models than on open-source ones?

* In Section 4.4, since the choice of the initial tool is critical, does Cross-Reflect perform better because it identifies the correct first tool with fewer trial steps?

* How does Cross-Reflect handle redundant tool invocation and logical inconsistencies compared to the baselines?

---

### Official Review · Reviewer_Tzbh · 2025-10-31

**Soundness:** 3
**Presentation:** 2
**Contribution:** 2
**Rating:** 2
**Confidence:** 3

**Summary:**

The paper introduces Cross-Reflect, a training-free framework for optimizing reasoning trajectories in multi-modal agents. The motivation is that fine-tuning small Vision-Language Models (VLMs) can be impractical due to computational and data constraints. Cross-Reflect aims to address this by jointly reflecting on reasoning actions and thoughts across multiple trajectories to enhance performance without further training.
The method yields notable improvements on visual question answering (VQA) benchmarks, reportedly up to +10% for proprietary models and +28% for open-source models over baseline methods.

**Strengths:**

- The paper proposes a novel and intuitive idea, a reflection-based, training-free optimization method for small VLMs.
- The structure and presentation are overall clear and logically organized.
- The authors provide ethics, reproducibility, and broader impact statements, demonstrating awareness of research responsibility.

**Weaknesses:**

- The proposed method resembles a best-of-N search over reasoning trajectories rather than a novel optimization mechanism.
- Although the paper claims reproducibility (see reproducibilty statement) and builds on an open-source framework (DSPy), the authors do not release code, which undermines the reproducibility claim.
- There is no discussion of the method’s limitations, e.g., potential computational cost or failure cases.
- Tables are difficult to read, especially Table 1, the font is too small.
- Similarly, Figure F is blurred.
- The authors claim that their training-free method outperforms fine-tuning in some cases and present comparison with OmniSearch to bakc up this statement.
However, the comparison with OmniSearch is somewhat misleading: at lines 365–366, the paper claims that Cross-Reflect + Qwen2.5-VL-7B surpasses OmniSearch. However, if OmniSearch uses an earlier backbone (e.g., Qwen-VL), then the comparison is unfair since Qwen2.5-VL is a much stronger model. A fair comparison would require both methods to use the same backbone. I highly encourage authors to add an experiment with the same backbone for fair comparison. Otherwise the claim cannot hold.

**Questions:**

- What exactly is meant by “action-thinking dual reflection” (line 148)? Please clarify how this differs from standard self-reflection or reasoning re-evaluation methods.
- How do you define “small VLMs” in this context?

---

### Official Review · Reviewer_5Bxf · 2025-11-01

**Soundness:** 2
**Presentation:** 2
**Contribution:** 2
**Rating:** 4
**Confidence:** 5

**Summary:**

This paper presents Cross-Reflect, a training-free framework for improving small vision-language models (VLMs) on knowledge-intensive visual question answering tasks. The approach extends DSPy to support multimodal inputs and introduces a reflection-guided mechanism that generates multiple reasoning trajectories, produces structured reflections to guide subsequent rollouts, and performs cross-trajectory selection to identify the most reliable answer. Experiments on InfoSeek, EncVQA, and Dyn-VQA demonstrate substantial improvements over baseline methods, with the framework achieving competitive performance compared to fine-tuned approaches.

**Strengths:**

The paper addresses an important problem: enabling small VLMs to perform complex multi-step reasoning without fine-tuning, which has significant practical implications for resource-constrained deployments. This motivation is clear and well-articulated throughout the manuscript.

The experiments are thorough, covering multiple benchmarks (InfoSeek, EncVQA, Dyn-VQA), both proprietary and open-source models, and include detailed ablations demonstrating the contribution of key components. The experimental design allows for fair comparison across different model families and baseline methods.

**Weaknesses:**

1. While the empirical results are strong, the core technical contributions appear incremental. The framework primarily combines existing concepts (ReAct-style reasoning, trajectory sampling, reflection mechanisms) rather than introducing fundamentally new techniques. The reflection mechanism, while structured and effective, resembles best-of-N sampling with LLM-based judging.

2. Although the authors claim several innovations (DSPy initialization, structured reasoning, trajectory-level reflection), the relationship to ReAct and what specifically makes this more than an engineered extension remains unclear. The heavy reliance on ReAct as both a baseline and component suggests the contribution may be more incremental than presented.

3. The paper provides limited insight into when and why Cross-Reflect fails, or how reflection might introduce new errors. The qualitative examples in the appendix are helpful but focus on success cases. More analysis of failure modes would strengthen the work and provide guidance for future improvements.

4. The paper does not adequately address known limitations of self-reflection in LLMs. While the authors distinguish their trajectory-level reflection from answer-level reflection, concerns about self-verification capabilities remain partially unaddressed. The references provided by reviewers regarding CoT and self-reflection limitations deserve more thorough engagement.

**Questions:**

1. The cross-model experiments (Planning: Qwen, Reflection: GPT-4o-mini achieving 37.4%) suggest that model alignment matters for reflection quality. Why does weak-model reflection sometimes hurt strong-model performance? This finding deserves deeper investigation as it has implications for the self-reflection paradigm.

2. How does the method scale with the number of trajectories? Is there a point of diminishing returns? The paper samples 3 trajectories but does not explore this design choice systematically.

3. Can you provide more details on when reflection successfully corrects errors versus when it fails or introduces new mistakes? What patterns distinguish successful reflection from unsuccessful attempts?

---

### Meta-Review · Area_Chair_8CN8 · 2026-01-09

**Summary:**

Reviewers raised critical concerns, including incremental technical contributions (combining existing concepts without novel techniques), lack of code undermining reproducibility, presentation flaws (unclear figures/tables, poor structure), insufficient experiments (unfair baselines, limited model evaluations), unaddressed failure modes/limitations, computational inefficiency, and ambiguous distinctions from existing methods (e.g., ReAct). No rebuttals are provided. These issues collectively justify rejection.

**Reviewer Concerns:**

No rebuttals are provided. Remaining concerns include incremental contributions, missing code/reproducibility, presentation issues, inadequate experiment design (unfair comparisons, limited baselines/models), unanalyzed failure modes/limitations, computational inefficiency, and unclear innovations vs. existing methods.

**Reviewer Scores:**

The authors do not write rebuttals, so reviewers would not have changed their scores.

---

### Decision · Program_Chairs · 2026-01-26

Reject